# Single-cell transcriptome analysis of epithelial, immune, and stromal signatures and interactions in human ovarian cancer

Chaochao Chai[1,2,3,7], Langchao Liang[1,2,3,7], Nanna S. Mikkelsen[4], Wei Wang[5], Wandong Zhao[1,2], Chengcheng Sun[1,2], Rasmus O. Bak [4], Hanbo Li [2,3], Lin Lin [4,6], Fei Wang [2,3,4,8 ✉] & Yonglun Luo [2,3,4,6,8 ✉]

A comprehensive investigation of ovarian cancer (OC) progression at the single-cell level is crucial for enhancing our understanding of the disease, as well as for the development of better diagnoses and treatments. Here, over half a million single-cell transcriptome data were collected from 84 OC patients across all clinical stages. Through integrative analysis, we identified heterogeneous epithelial-immune-stromal cellular compartments and their interactions in the OC microenvironment. The epithelial cells displayed clinical subtype features with functional variance. A significant increase in distinct T cell subtypes was identified including Tregs and CD8+ exhausted T cells from stage IC2. Additionally, we discovered antigen-presenting cancer-associated fibroblasts (CAFs), with myofibroblastic CAFs (myCAFs) exhibiting enriched extracellular matrix (ECM) functionality linked to tumor progression at stage IC2. Furthermore, the NECTIN2-TIGIT ligand-receptor pair was identified to mediate T cells communicating with epithelial, fibroblast, endothelial, and other cell types. Knock-out of *NECTIN2* using CRISPR/Cas9 inhibited ovarian cancer cell (SKOV3) proliferation, and increased T cell proliferation when co-cultured. These findings shed light on the cellular compartments and functional aspects of OC, providing insights into the molecular mechanisms underlying stage IC2 and potential therapeutic strategies for OC.

[1] College of Life Sciences, University of Chinese Academy of Sciences, Beijing 10049, China. [2] Lars Bolund Institute of Regenerative Medicine Qingdao-Europe Advanced Institute for LifeScience, BGI Research, Qingdao 266555, China. [3] BGI Research, Shenzhen 518083, China. [4] Department of Biomedicine, Aarhus University, Aarhus, Denmark. [5] Department of Obstetrics and Gynecology, Tongji Hospital, Tongji Medical College, Huazhong University of Science and Technology, Wuhan, China. [6] Steno Diabetes Center Aarhus, Aarhus University Hospital, Aarhus, Denmark. [7]These authors contributed equally: Chaochao Chai, Langchao Liang. [8]These authors jointly supervised this work: Fei Wang, Yonglun Luo. ✉email: wangfei@biomed.au.dk; alun@biomed.au.dk

Ovarian cancer (OC) is a malignant gynecological tumor with a high mortality rate[1]. In 2022, there were 57,090 new OC cases reported in China, which is twice of cases reported in the United States in the same year[2]. In 2023, OC is estimated to account for 5% of cancer-related deaths among women in the United States[3]. High-grade serous ovarian cancer (HGSOC) is the most common histological subtype of OC, accounting for 75% of all OC cases. The five-year survival rate of OC is higher in the early stages, unfortunately, approximately 75% of the patients are diagnosed in the later stages[4]. Advanced OC patients typically have more metastatic sites, resulting in a higher cancer recurrence rate and a lower five-year survival rate[5]. Currently, screening methods for OC, such as detecting serum CA125 concentration and ultrasonography, have a high rate of false negatives and false positives in the general population and are not universally applicable[6]. However, some new biomarkers have been reported in combination with CA125, showing a certain effect in malignancy risk algorithms[7]. Thus, investigating the cellular and molecular mechanisms of OC in all clinical stages can advance our understanding of the pathogenesis of OC, help discovering new biomarkers, and support early screening and treatment of OC.

OC is a malignancy with an unclear etiology and has intratumoral heterogeneity. Single-cell RNA sequencing (scRNA-seq) has been widely used to dissect the cellular compositions and heterogeneity in the tumor microenvironment (TME) in various types of cancer[8–10]. For OC, Xu et al. recently reported the complex changes in TME using scRNA-seq, including the decreased attraction of macrophages to immune cells and the variance in the interaction between cancer-associated fibroblasts (CAFs) and cancer cells[11]. Understanding the cellular and molecular changes during OC progression is beneficial for discovering the dynamic changes of OC, particularly for identifying potential biomarkers for diagnosis and treatment[12]. OC has a complex TME and exhibits higher intratumoral heterogeneity compared to other gynecological tumors[13,14]. Single-cell studies have shown that cancer cell subsets have both patient-specific features[14,15], and functional features[16], but the functional differences in epithelial cells in OC stages are still unclear. CAF is an important component of the TME in OC and plays a vital role in tumor pathogenesis[17]. The interaction between cancer cells and primary myCAF can promote the epithelial-mesenchymal transition (EMT) process in tumors[11], and co-activation of certain signaling pathways in CAFs and cancer cells in OC ascites promotes drug resistance[9,18]. However, the variety of interactions between CAFs and other cell types during OC progression remains poorly understood.

OC is characterized as an immunogenic tumor, showcasing a spontaneous antitumor immune response. Tumor-infiltrating lymphocytes have been demonstrated to strongly correlate with a favorable prognosis in the OC microenvironment[19]. Anadonet et al. explored the distribution of CD8 + T cells in OC and found that effector T cells constitute a small portion of CD8 + T cells, with tumor-infiltrating lymphocytes mainly acting as bystanders in the TME[20]. T-cell exhaustion is characterized by the stepwise and progressive loss of T-cell effector functions and homeostatic self-renewal capacity during tumor outgrowth[21–23]. Understanding the properties and pathways to T-cell exhaustion has crucial implications for the success of checkpoint inhibitors and adoptive T-cell therapies. A few studies showed that T-cell exhaustion features could be a predictor for the prognosis and treatment of OC patients[19,24]. However, the features and dynamic lineage changes of T cell exhaustion during OC progression are still unclear, making it necessary to further explore the gene expression differences and the cellular interaction relationships during the OC progression.

In this study, we report the integration of single-cell transcriptomic datasets of OC encompassing all clinical stages, comprising data from 84 OC patients with a total of 505,102 single cells. We performed an integration analysis and identified eight major cell types in OC. We identity many subtypes of epithelial cells, T cells, and fibroblasts in OC. The epithelial cell subtypes exhibited distinct cell functions, copy number variation (CNV), and gene expression changes during OC progression. Additionally, we thoroughly classified T cell subtypes, investigated the differentiation trajectory of CD8 + T cell subtypes in OC, and explored the changes in the proportion of T cell subsets during OC progression. Among the different cell types, fibroblasts exhibited the most interaction with epithelial cells and T cells. Interestingly, we observed a clear increase in the expression of ECM-related genes in myCAF along the progression of the tumor stages. Finally, we compared the changes in cellular communication probability from fibroblast to epithelial cells, as well as fibroblast to T cells across different stages of OC. Our findings revealed an increasing interaction of the COLLAGEN and LAMININ pathways in the fibroblast-to-epithelial cell communication as the disease advances. Furthermore, most cell types exhibited immunosuppressive effects on T cells through the NECTIN2-TIGIT ligand-receptor pair interaction pathway.

## Results

**An integrated single-cell transcriptome atlas of human ovarian cancer.** Several previous studies have reported the analysis of cellular heterogeneity and functions in human OC with scRNA-seq, but differ in terms of clinical stages. We first sought to generate an integrated and curated single-cell transcriptome atlas, which will benefit the systematic analysis of the cellular compositions and heterogeneity of OC. We collected and reprocessed 109 scRNA-seq datasets (Supplementary Data 1.1) and filtered out the low-quality data using the methods described (see Methods section). Ultimately, we obtained a total of 505,102 high-quality cells from 9 OC clinical stages for subsequent analysis (Fig. 1a, Table 1, *detailed in methods*). Initially, we performed an integration analysis on different samples, tissues, and clinical stages of OC using harmony[25] (Fig. 1b, Supplementary Fig. 1a). Based on expression of canonical markers, we identified eight major cell types, including B cells (*CD79A, IGHM*), endothelial cells (*VWF, CLDN5*), epithelial cells (*KRT18, WFDC2*), fibroblasts (*DCN, COL1A1*), mast cells (*KIT, MS4A2*), myeloid cells (*LYZ, APOE*), pericytes (*TAGLN, RGS5*), and T cells (*PTPRC, CD3D*) (Fig. 1c, Supplementary Fig. 1b). To validate the accuracy of the cell type annotation, we performed differentially expressed genes (DEGs) analysis for each cell type and conducted Gene Ontology (GO) enrichment analysis (Fig. 1d, Supplementary Data 1.2, 1.3), which further confirmed correct cell type annotations.

We then investigated the proportions of each major cell type in different source studies and tissues. CD45-positive immune cells from the OvC09[26] study mainly comprised T cells, myeloid cells, and B cells. In line with this, the OvC11[20] study focused on T cells, and over 90% of our annotation results were also identified as T cells (Fig. 1e), further validating the accuracy of our cell-type annotation. When considering different metastatic tumor foci, we observed that the normal ovarium primarily consisted of pericytes. However, in ovarium tumors, the proportion of epithelial cells increased, and this proportion was further elevated in relapse tumors. Notably, the fractions of both epithelial cells and T cells were increased in each metastasis compared to their respective normal tissues, such as omentum versus (vs.) normal omentum, and peritoneum vs. normal peritoneum. These findings suggest that the immune

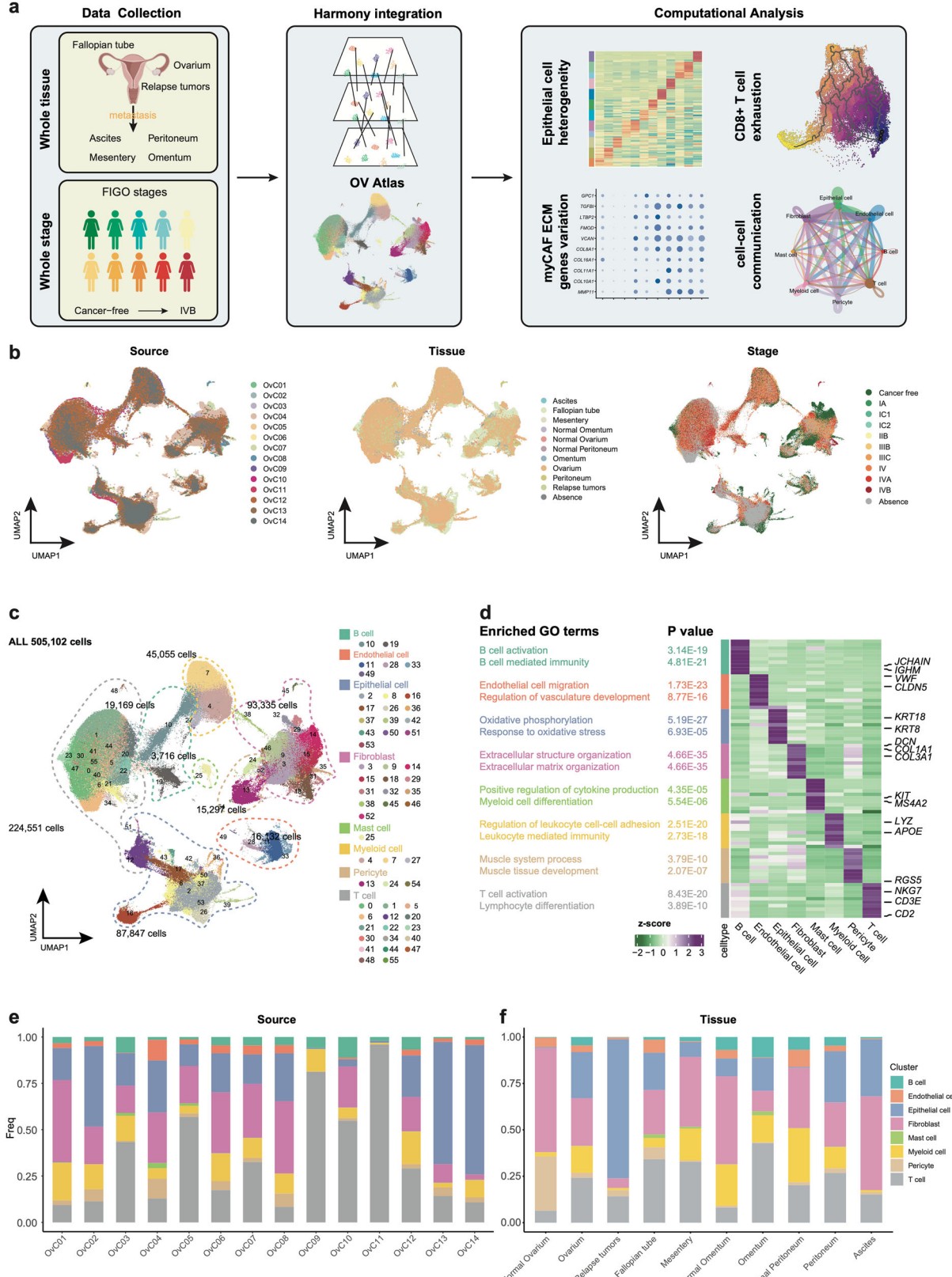

**Fig. 1 Dissection of ovarian cancers with scRNA-seq. a** Workflow depicting the collection and processing of ovarian cancer for scRNA-seq. Illustration of female reproductive system and female icons were created with BioRender.com with licensed for use in journal publications. **b** The UMAP plot shows cell literature source, tissue and stage by color. **c** The UMAP plot demonstrates the major cell types in ovarian tumor and metastatic tumor foci. **d** The expression heatmap of top10 marker genes for major cell type and the GO enrichment term of marker genes. **e** Stacked bar graph of the proportion of major cell types in each literature source in scRNA-seq profiles. **f** Stacked bar graph of the proportion of major cell types in each metastatic tumor foci in scRNA-seq profiles.

**Table 1 Overview of the data used in this study.**

| Source_ID | Cells | Number of patients | Stage (FIGO) | Tissue (metastatic tumor foci) | Histological subtype |
|---|---|---|---|---|---|
| OvC01[27] | 30317 | 3 | NA | Ovarium, Ascites | HGSOC |
| OvC02[13] | 13950 | 2 | IIB, IIC | Ovarium | HGSOC |
| OvC03[9] | 51786 | 11 | IIIC, IVA, IVB | Peritoneum, Omentum, Mesentery, Ovarium | HGSOC |
| OvC04[60] | 59738 | 4 | Healthy | Fallopian tube | fallopian tubes cancer-free |
| OvC05[61] | 62755 | 12 | Cancer-free | Fallopian tube | fallopian tubes cancer-free |
| OvC06[62] | 45114 | 10 | IIIC, IVB, IA | Peritoneum, Omentum, Ovarium | 4HGSOC + 1HGSOC_clear cell carcinoma (mix) |
| OvC07[11] | 59661 | 12 | Cancer-free, IIIB, IIB, IC2, IIC | Ovarium | HGSOC |
| OvC08[63] | 18403 | 7 | IIIC, IVB, IC1 | Peritoneum, Omentum, Ovarium | 7HGSOC + 1HGSOC_clear cell carcinoma (mix) |
| OvC09[26] | 26703 | 6 | IIIC | Ovarium | HGSOC |
| OvC10[64] | 3827 | 1 | NA | NA | HGSOC |
| OvC11[20] | 101651 | 9 | NA | NA | HGSOC |
| OvC12[16] | 50396 | 5 | III-IV | Ovarium | HGSOC |
| OvC13[65] | 31969 | 8 | NA | Ovarium, Peritoneum, Relapse tumors | 2HGSOC + 4LGSOC + SOC+Endometrioid |
| OvC14[66] | 7554 | 3 | NA | Ovarium | HGSOC |

*LGSOC Low-grade serous ovarian cancer.*

microenvironment of metastatic carcinoma foci resembles that of primary carcinomas (ovarium vs. normal ovarium) in terms of changes in the proportion of major cell types. In ascites, we observed a high number of epithelial cells and fibroblasts. This data was derived from the OvC01[27] dataset, in which the original study had adopted a strategy of eliminating CD45+ immune cells (Fig. 1f).

In summary, our analysis integrated scRNA-seq datasets of OC, particularly HGSOC. We encompassed primary tumors, relapse tumors, fallopian tubes, and metastatic carcinoma foci, covering all stages from healthy, cancer-free individuals to stage I–IV patients, totaling 84 patients. Additionally, we have created an interactive website (https://dreamapp.biomed.au.dk/OvaryCancer_DB/) for the visualization of the datasets.

**Epithelial cell subtypes and clinical stage-related functional differences in ovarian cancer**. To identify molecular signatures of epithelial cells in OC, we subset and performed single-cell analysis on a total of 87,847 epithelial cells. Using consensus non-negative matrix factorization[28], we generated transcriptional gene modules and divided the cell clusters based on the gene expression matrix. We assessed the stability and error of the module quantity and ultimately identified the optimal 9 gene modules to reduce the data dimension (Supplementary Fig. 2a). The epithelial cells were categorized into a total of 12 clusters (Epi0-Epi11), and we named functional clusters by DEGs (Supplementary Data 2.1). Epithelial cells exhibit a large degree of tissue heterogeneity and demonstrate a better integration effect between OC stages in UMAP dimensions (Fig. 2a, Supplementary Fig. 2b–d). Each of the 12 clusters corresponded to one or more functional modules. Clusters such as Epi0, Epi3, Epi4, and Epi7 corresponded to a single module, while clusters Epi1 and Epi2 corresponded to multiple modules (Fig. 2b). Each cluster exhibited specific DEGs, with clusters corresponding to the same module, such as Epi1 and Epi7, displaying relatively consistent DEGs (Fig. 2c). We performed GO enrichment analysis on the top 200 contributing genes of each gene module. Tumor cell function gene sets, which are provided by the CancerSEA database[29], were scored with gene set variation analysis (GSVA) using the gene contribution scores of each gene module (Supplementary Data 2.2, 2.3). Modules 3, 6, and 9 represented gene modules of normal epithelial cells, enriched in functions such as inflammation, cilium formation, and steroid metabolism, which aligned with the tissue source of the corresponding clusters. Modules 2, 4, and 7 are enriched in basic cell functions and related signaling pathways such as mitochondrion-related functions and ATP biosynthetic processes. Module 5 exhibited a higher enrichment score for hypoxia and metastasis gene sets, while module 8 displayed higher scores for replication-related gene sets and proliferation gene sets. Module 1 was present in both cancer-free fallopian tube epithelium and tumor cells, displaying higher scores of EMT and invasion gene sets, and primarily enriched in ECM-related signaling pathways (Fig. 2b, c, Supplementary Fig. 2b–d, e-f, Table 2).

To investigate the correlation between CNV in epithelial cells and cancer progression, we randomly selected 30,000 epithelial cells for *inferCNV* analysis (https://github.com/broadinstitute/inferCNV). We compared the CNV profiles of epithelial cells across different stages, and the results revealed significant ($P < 0.001$, Wilcoxon rank-sum test) CNV differences between normal epithelial cells and tumor cells. While there was only a slight difference in CNV between stage I and normal tissue, stages II to IV exhibited higher levels of CNV compared to stage I. This observation suggests that OC stage I has a lower degree of malignancy compared to the later stages (Fig. 2d, e, Supplementary Fig. 2g). Additionally, we evaluated the SCENT cell

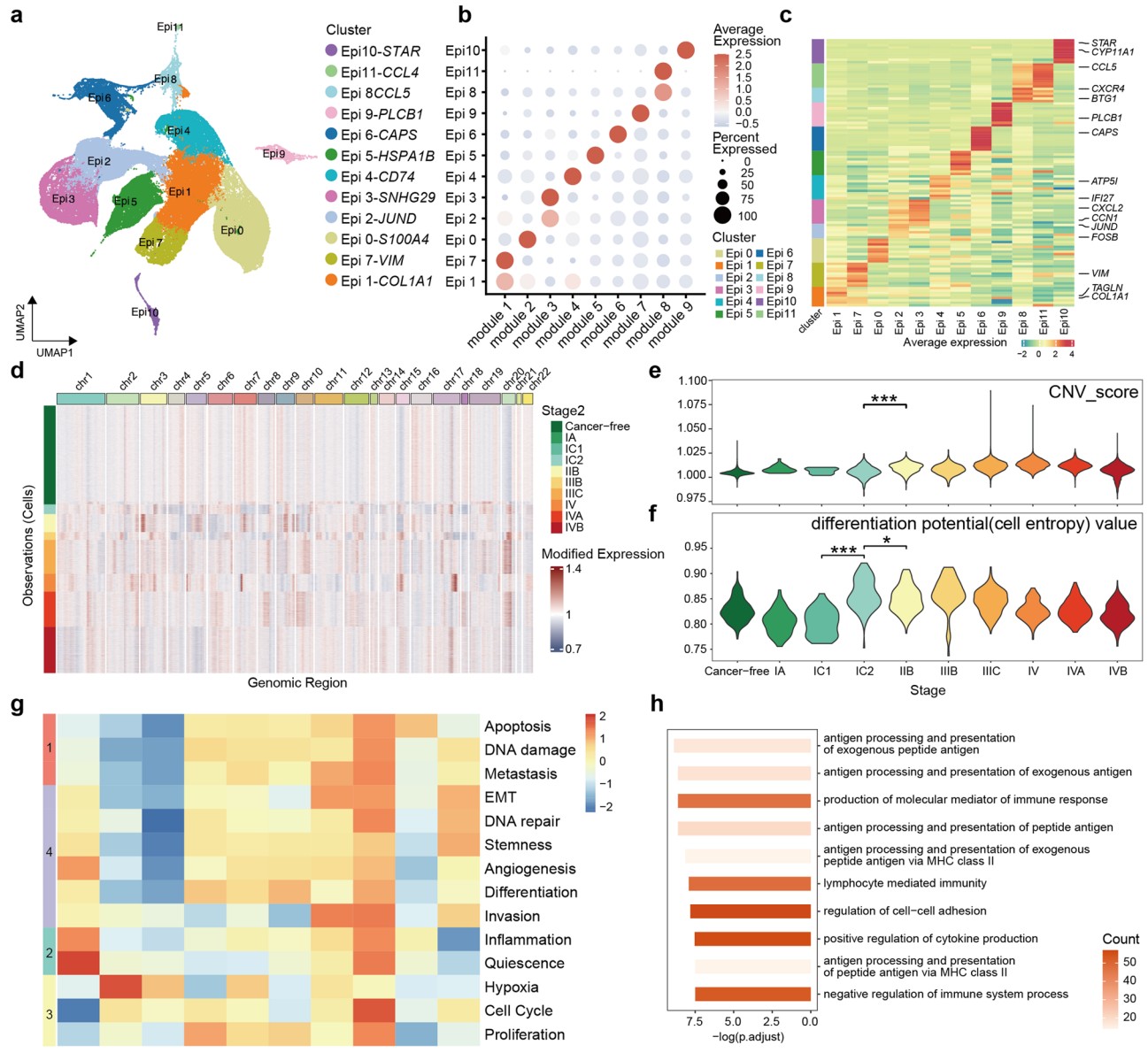

**Fig. 2 Epithelial cell subtypes and clinical stage-related functional differences. a** The UMAP plot shows epithelial cell sub cluster by color. **b** Dot plot shows gene module contribution of epithelial cell sub cluster. **c** The expression heatmap of top10 marker genes for epithelial cell sub cluster. **d** Heatmaps shows large-scale CNVs for individual epithelial cells. **e** The violin plot shows the cell entropy value of epithelial cells in each stage (Total 23,564 single cell). *$P < 0.05$; **$P < 0.01$; ***$P < 0.001$, ****$P < 0.0001$. **f** The violin plot shows the CNV score of epithelial cells in each stage (Total 2,488 single cells). **g** The heatmap shows the tumor cell function gene sets score of each stage. **h** The bar chart shows the GO enrichment term of up regulated genes in stage IC2.

differentiation potential of epithelial cells by randomly estimating the entropy value of 3000 cells. During the transition from normal tissue to stage I and subsequently to stages II to IV, the cell entropy value showed a significant increase ($P < 0.001$) at stage IC2 and gradually decreased in the later stages (Fig. 2f).

To compare the differences in tumor cell functions in the OC stages, we calculated the scores of tumor cell function gene sets for each stage. We observed 4 distinct trends across 14 gene sets. Trend 1 showed low function scores in stages before stage IC2 and high scores in later stages. Trend 2 represented high function scores in cancer-free cells and low scores in tumor cells. Trend 3 was the opposite of trend 2, indicating that compared to cancer-free cells, cancer cells exhibited enhanced hypoxia and proliferation functions, and cancer-free cells remained relatively quiescent and exhibited inflammation. During the transition from stage IC1

to IC2, tumor cells exhibited elevated apoptosis, DNA damage, and metastasis functions (Fig. 2g). The percentage differences in cell clusters during stage progression were consistent with functional scoring results. Cluster Epi3, associated with inflammation, was predominantly presented in the cancer-free cell group, while cluster Epi5, consisting of proliferating cells, was exclusively found in the cancer stages. Cluster Epi4 enriched for apoptosis and metastasis, appeared in stages after stage IC1 (Fig. 2g, Supplementary Fig. 2d).

Furthermore, we used *Mfuzz*[30] to assess the gene expression differences in epithelial cells during OC stage progression (Supplementary Fig. 2h). Among the 20 gene change trend clusters, clusters 4, 5, 10, and 11 exhibited an upward trend from stage IC1 to IC2. Notably, these gene clusters included the OC early screening gene *MUC16* (CA-125), *WFDC2* (HE4), and

**Table 2 Summary of the function of each epithelial cell subgroup.**

| Cluster | module | function |
|---|---|---|
| Epi0 | module 1, module 2, module 4 | EMT, Invasion, basic cell functions |
| Epi1 | module 1 | EMT, Invasion |
| Epi2 | module 1, module 3 | EMT, Invasion, inflammation |
| Epi3 | module 3 | Inflammation |
| Epi4 | module 4 | basic cell functions |
| Epi5 | module 5 | Hypoxia, Metastasis |
| Epi6 | module 6 | Cilium organization |
| Epi7 | module 1 | EMT, Invasion |
| Epi8 | module 8 | Proliferation |
| Epi9 | module 7 | basic cell functions |
| Epi10 | module 9 | Steroid metabolic process |
| Epi11 | module 8 | Proliferation |

*PAX8* (Supplementary Data 2.4). These genes were enriched in signaling pathways related to the major histocompatibility complex class-II (MHC-II) antigen presentation pathway. Increased expression of tumor-enriched MHC-II has been associated with a favorable prognosis and increased numbers of tumor-infiltrating CD8 + T lymphocytes[31,32]. However, it has also been reported that tumor cell MHC-II molecules may engage with ligands to induce immunosuppression[33] (Fig. 2h, Supplementary Data 2.5). These results suggest that the transition from stage I to stage II may represent a critical time node in the progression of OC, wherein CNV levels, differentiation potential, and cell function of cancer cells undergo robust changes. The genes exhibited an upward trend between stage IC1 and IC2 including previously reported OC screening genes, while the remaining genes showing an uptrend represented potential candidates for OC screening.

**The trajectory of T cell subtypes during ovarian cancer progression.** Immunotherapy targeting inhibitory immune checkpoints in immune cells has shown remarkable efficacy in treating various solid tumors[34,35]. T cells, as the crucial immune cells, play a pivotal role in tumor development and treatment. To investigate the changes in T cells across different stages of HGSOC, we performed a detailed classification and annotation of T cell subtypes and presented the proportions of T cell subtypes from different sources, clinical stages, tissues, and patients (Supplementary Fig. 3a). Specifically, CD4 + T cells (*CD4*), CD4+ regulatory T cells (Tregs) (*CD4, IL2RA, FOXP3*), CD8+ naive T cells (*CD8A, CCR7, SELL, TCF7*), CD8+ memory T cells (*CD8A, GPR183, IL7R*), CD8+ effector T cells (*CD8A, GNLY, GZMB2, PRF1*), CD8+ exhausted T cells (*CD8A, PDCD1, TIGIT, HAVCR2, CTLA4*), and CD8+ proliferating T cells (*CD8A, TOP2A, MKI67*) were identified (Fig. 3a, b). Interestingly, we observed that CD8+ proliferating T cells exhibited both proliferation characteristics and exhaustion features. This finding is consistent with the study by Carmen et al.[20] and has also been reported in renal carcinoma[8]. The UMAP plot demonstrated that CD8+ proliferating T cells were positioned at the end, suggesting that they may represent the final lineage stage of CD8 + T cells. To validate this observation, we performed a pseudo-time analysis of CD8 + T cells, revealing that CD8+ proliferating T cells were derived from CD8+ naive T cells through two potential routes: 1. CD8+ naive T cells → CD8+ memory T cells → CD8+ effector T cells → CD8+ exhausted T cells → CD8+ proliferating T cells; 2. CD8+ naive T cells → CD8+ exhausted T cells → CD8+ proliferating T cells (Fig. 3c). It is also demonstrated by the expression changes of the classical marker genes in CD8 + T cell subtypes along the pseudo-time trajectory. Notably, the naive genes for T cells are highly expressed in the early stage, effector genes are highly expressed at the middle stage, exhaustion genes are highly expressed at the middle and late stages, and pro-liferation genes are highly expressed at the end of the pseudo-time trajectory (Fig. 3d). Interestingly, our findings differ from those of Carmen et al., who observed that CD8+ proliferating T cells precede CD8+ exhausted T cells[20]. Further experimental verification is required to reconcile these discrepancies.

**CD8+ exhausted T cells and Tregs increased from stage IC1 to IC2.** We then examined the cell proportions of different T cell subtypes in OC at different stages (Fig. 3e). Notably, data from the OvC09, OvC10, and OvC11 studies were excluded here, as these three studies primarily focused on some subtypes of T cells and would impact the overall T cell proportions. We observed that there was clear difference in the proportion of T cell subtypes between the OC stages of cancer-free, stage IA, and stage IC1. However, from stage IC2 onwards, there was a clear increase in the proportions of CD8+ exhausted T cells and Tregs, while the proportion of CD8+ naive T cells was decreased. Furthermore, at the molecular level, the expression of classical markers associated with exhaustion and Tregs was increased after stage IC2, while classical naive markers were decreased (Fig. 3e). We further validated these findings by using the Tregs differentiation and exhaustion gene sets scoring, which showed the increased gene scoring for Tregs and exhausted T cells from stage IC2 onwards, consistent with cellular proportion and molecular features (Supplementary Fig. 3b, c, Supplementary Data 3.1, 3.2). Patients with cancer-free, stage IA, and stage IC1 were divided into group1, and those with stage IC2 onwards were grouped into group2. Statistical significance tests were performed on the cell proportions of patients in both groups, confirming the significant changes in cell proportions for Tregs ($P < 0.0001$), CD8+ exhausted T cells ($P < 0.05$), CD8+ naïve T cells ($P < 0.05$), and CD8+ proliferating T cells ($P < 0.05$) between these two groups (Fig. 3f). We also performed clustering analysis based on the temporal patterns of genes with high variation, which further confirmed these findings. Moreover, we identified additional genes that show prominent expression changes from stage IC1 to stage IC2 (Supplementary Fig. 3d, Supplementary Data 3.3). These results suggest that the obvious features of the immunosuppressive TME with the accelerated T cell exhaustion and the involvement of Tregs after stage IC2 of HGSOC, which indicated the potential contributions to cancer progression and metastasis at the late stage of HGSOC. We also examined the differences in T cell subtypes between primary tumors, metastatic tumor sites, and their corresponding normal tissue. We observed that the proportion of CD8+ exhausted T cells and Tregs was increased in the tumor samples, while CD8+ effector T cells and CD8+ memory T cells were decreased in the tumor compared to normal tissue (Supplementary Fig. 3e). These findings were consistent in both primary and metastatic tumors. Additionally, it is challenging to diagnose HGSOC in the early stages, and is prone to abdominal disseminated metastasis after stage II[36]. These changes observed at the single-cell and molecular levels in stage IC2 imply the potential applications for the early clinical detection of HGSOC.

**Antigen-presenting CAF was identified in ovarian cancers.** We next sought to investigate the functional phenotypes of CAFs, which are crucial for tumor occurrence and progression. We performed cell receptor and ligand interaction analysis and observed that fibroblasts had a higher number and intensive interactions with T cells and epithelial cells. This finding indicates that fibroblasts played an important role in TME (Fig. 4a). To further reveal the fibroblasts phenotypes, we performed high-

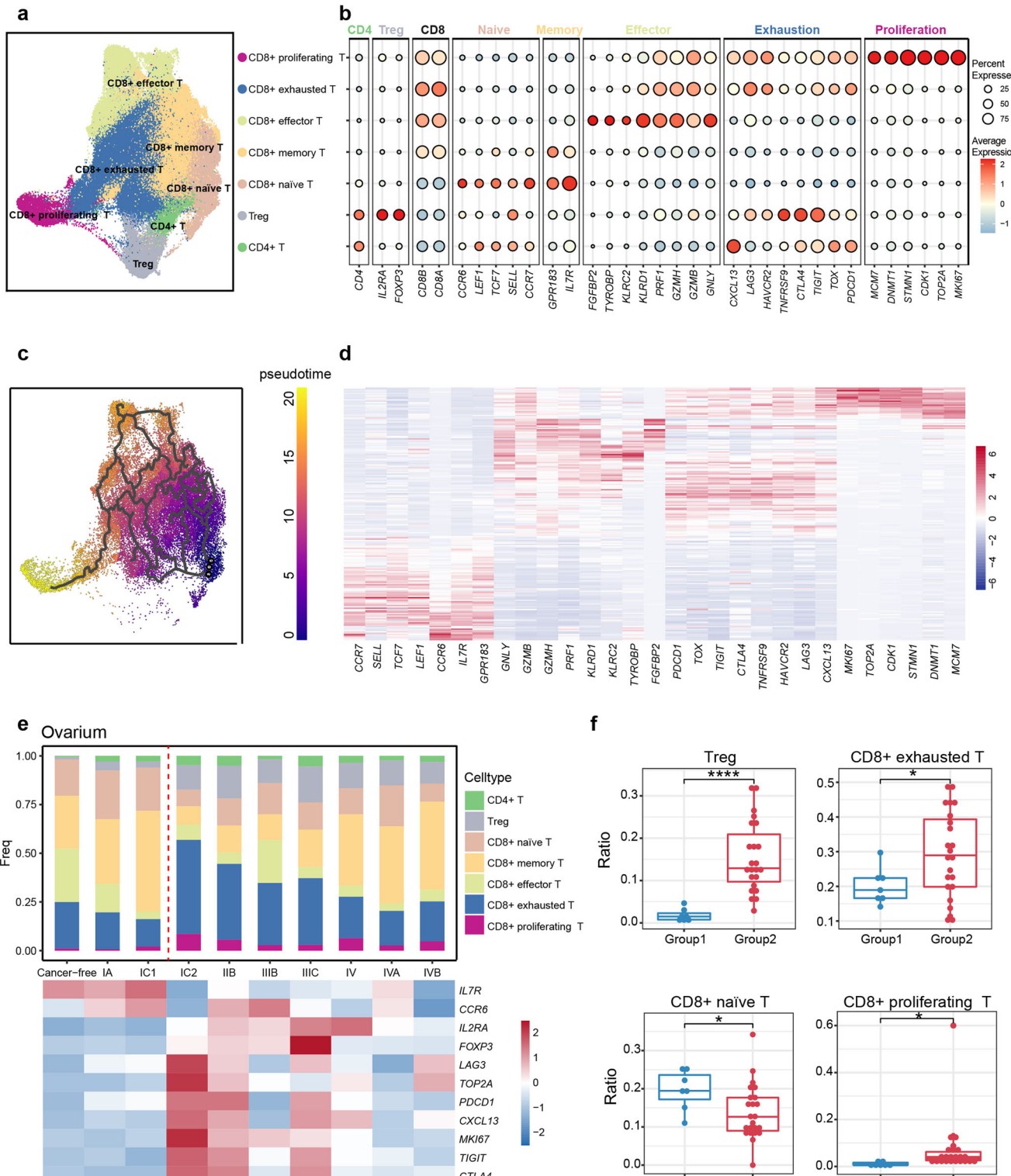

**Fig. 3 The pseudo-time trajectory of T cell subtypes and cell proportion change. a** The UMAP plot demonstrates the T cell subtypes in ovarian tumor and metastatic tumor foci. **b** Dot plot shows the average expression level and expression percentage of classic marker genes in T cell subtypes. **c** The UMAP plot shows the pseudo-time differentiation trajectory of CD8 + T cell subtypes with colors indicates pseudo-time. **d** The heat map shows the variation of CD8 + T cell subtype classical marker gene expression with pseudo-time. **e** Changes in the proportion of T cell subtypes and marker gene expression of naïve, exhaustion and Treg with stage in ovarian tumors. **f** The boxplot shows the difference in the cell proportion of Treg, CD8+ exhausted T cells, CD8+ naïve T cells and CD8+ proliferating T cells in group1 and group2. Each point indicates one patient (Total 30 patients, including 7 patients in group1, 23 patients in group2). *P < 0.05; **P < 0.01; ***P < 0.001, ****P < 0.0001.

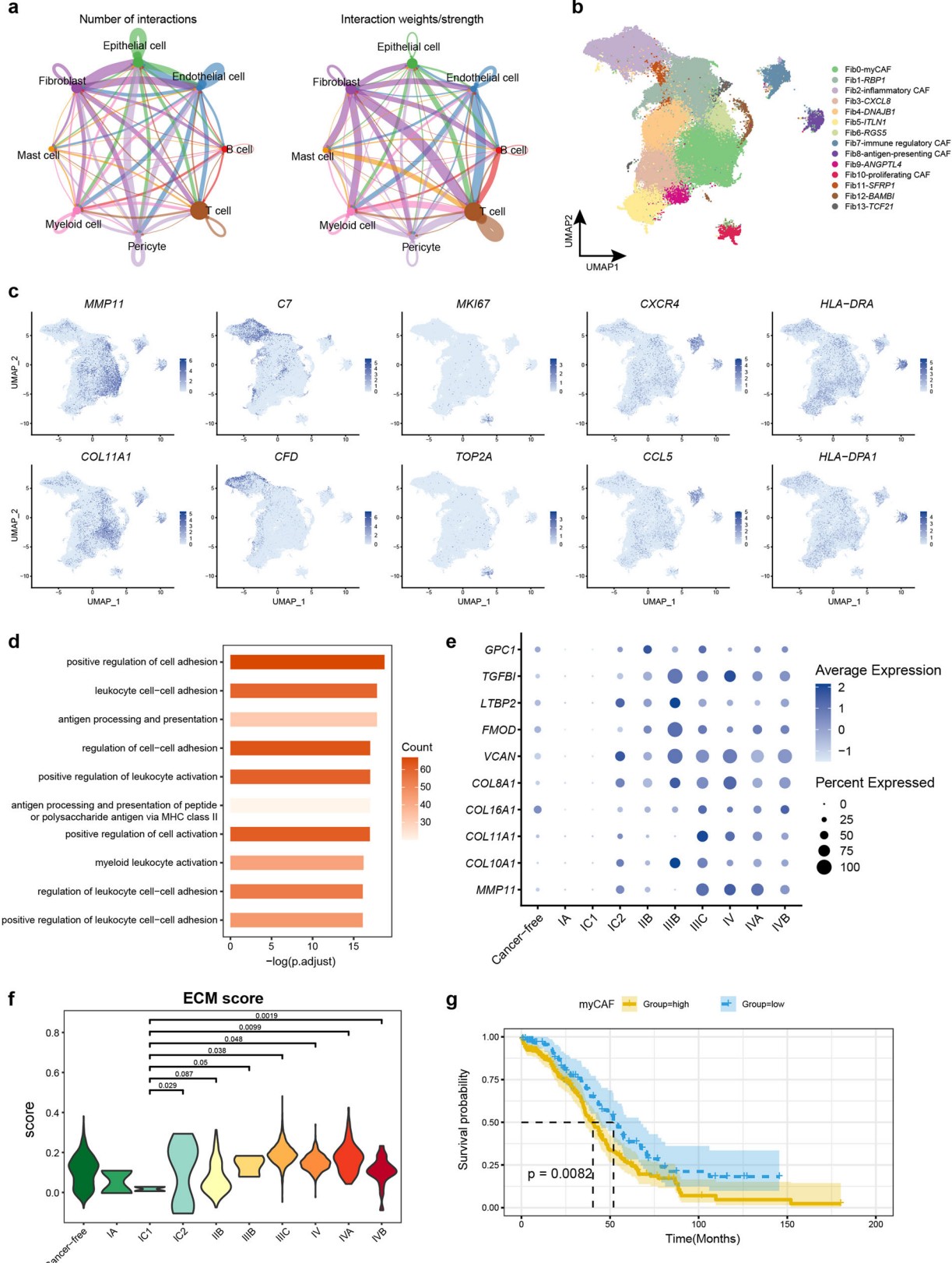

**Fig. 4 Antigen-presenting CAF was identified and enhanced ECM function in myCAF from stage IC2. a** The number and strength of cell interactions among the major cell types of ovarian cancer. **b** The UMAP plot shows the fibroblast sub-clusters in ovarian tumor and metastatic tumor foci. **c** The UMAP plot color coded for the expression of marker genes for the fibroblast subtypes. **d** The bar chart shows the GO enrichment term of marker genes in Fib8. **e** The dot plot shows variation of ECM gene expression in myCAF with stage. **f** The violin plot shows change of ECM score in myCAF with stage (Total 7,438 single cell). *$P < 0.05$; **$P < 0.01$; ***$P < 0.001$, ****$P < 0.0001$. **g** Kaplan–Meier OS curves of patients with TCGA HGSOC grouped by the top10 marker gene of myCAF. *P* values were calculated by a log rank test.

resolution clustering for fibroblasts and identified myCAF (Fib0: *MMP11*, *COL11A1*), inflammatory CAF (Fib2: *C7*, *CFD*, *DPT*), proliferating CAF (Fib10: *MKI67*, *TOP2A*), immune regulatory CAF (Fib7: *CXCR4*, *CCL5*), and antigen-presenting CAF (Fib8, *HLA-DRA*, *HLA-DRB*) (Fig. 4b, c). Notably, the antigen-presenting CAF was not identified by previous studies conducting the scRNA-seq of OC samples, although this cell type has been found in human lungs and livers and mouse pancreas and mammary glands[10,37–40]. Our integrative strategy of this large amount of scRNA-seq data helps with revealing the rare fibroblast phenotypes. Antigen-presenting CAF were found in various metastatic tumor foci (Supplementary Fig. 4a), which exhibited high expression of *HLA-DRA*, *HLA-DRB*, and other MHC genes (Fig. 4c, Supplementary Data 4.1), potentially contributing to anti-tumor immunity. In line with this, GO enrichment analysis of marker genes in Fib8 showed the enrichment of "antigen processing and presentation", "antigen processing and presentation of peptide or polysaccharide antigen via MHC class II", and "antigen processing and presentation of peptide antigen", which further validated Fib8 as the antigen-presenting CAF (Fig. 4d, Supplementary Data 4.2).

**Enhanced ECM function in myCAF from stage IC2.** MyCAFs are a class of CAFs with strong ECM deposition characteristics and are the main proportion of CAF in OC tumors (Fig. 4b, c). Interestingly, we observed an increase in the expression of ECM genes from stage IC2 (Fig. 4e), indicating a potential association between increased tumor cell growth/metastasis and the CAF activation and ECM formation from myCAFs. To further validate this finding, we calculated the ECM score using all genes involved in the ECM process (GO: 0031012) and observed a significant increase in ECM score from stage IC2 ($P < 0.05$) (Fig. 4f). Moreover, the fraction of myCAF was clearly elevated in each metastatic tumor foci compared to their corresponding normal tissues, such as omentum vs. normal omentum, and peritoneum vs. normal peritoneum (Supplementary Fig. 4a). Notably, the proportion of myCAF also increased clearly after stage IC2 (Supplementary Fig. 4b). To explore the potential of myCAF genes as prognostic biomarkers for HGSOC, we calculated the GSVA scores for the top 10 myCAF marker genes and conducted survival analysis using TCGA HGSOC data. Strikingly, we found that there was a significant correlation (long-rank test, $P$ value = 0.0082) between the high expression of these top marker genes and the poor prognosis of patients (Fig. 4g), indicating their potential as therapeutic targets in OC.

**Interactome between fibroblasts, epithelial cells, and T cells over OC progression.** The large number of ligand-receptor pairs between fibroblasts and T cells/epithelial cells indicate a high probability and importance of cellular communications between cell types in OC progression (Fig. 4a). This prompts us to explore the ligand-receptor pairs involved in the OC microenvironment, immunosuppression, tumor cell growth, and metastasis. We performed ligand-receptor interaction analysis for cells in five clinical stages of primary OC (cancer-free, stage I, stage II, stage III, and stage IV). This analysis revealed an increase in both numbers and intensity of cell interactions from cancer-free to stage I, which however gradually decreased over later tumor progression (Fig. 5a). These findings suggest that the increased cell-cell interactions during the early stage of cancer progression indicate the strong cellular and tissue remodeling within the establishment of TME, whereas immunosuppression (reduced cellular interactome, e.g., immune suppression) is essential for the advanced progression of OCs.

We investigated the interactions of fibroblasts to epithelial cells and fibroblasts to T cells. Our results showed that the extracellular matrix receptor interactions with COLLAGEN and LAMININ pathways exhibited the largest number of interactions (Fig. 5b, c, Supplementary Data 5.1). Both collagen and laminin have been reported to be associated with promoting OC metastasis[41–43]. The ligands of the COLLAGEN pathway primarily include collagen 1, collagen 4, and collagen 6, while the receptors were mainly integrin, SDC4, and CD44. The probability of cell communication was strongly enhanced in the progressive tumor stages compared with cancer-free fibroblasts to epithelial cells, but there were no evident changes for fibroblasts to T cells. Laminin belongs to the extracellular matrix glycoprotein family and is the main non-collagenous component of the basement membrane[44]. Its main receptors are integrin, DAG1, and CD44. In fibroblasts to epithelial cells interactions, various LAMININ molecules play a key role in different stages of tumor development, and the Laminin-CD44 interactions in fibroblasts to T cells exhibited a clear downward trend with tumor development. CD44 is usually expressed by activated T cells and is essential for the potency of T cell effector function[45]. These results suggested that fibroblast cells may interact with epithelial cells through COLLAGEN and LAMININ pathways in the progressive tumor stages, leading to tumor cell metastasis. In addition, fibroblast cells also regulate T cells through laminin-CD44, making T cell activity and efficacy decrease with the tumor progression.

After that, we focused on identifying the specific cell types within the OC TME that express immune checkpoint ligand genes, leading to the inhibition of T cell functions. Thus, we examined the immune checkpoint-related ligand-receptor interactions, and TIGIT stood out as the most obvious interactome. PVR-TIGIT, NECTIN2-TIGIT, and NECTIN3-TIGIT were not expressed in the cancer-free stage, but the obvious cell interaction appeared from stage I, indicating that immunosuppression had already appeared at the beginning of stage I. Additionally, TIGIT ligands are produced by various cell types, including endothelial cells, epithelial cells, fibroblasts, myeloid cells, and pericytes. Therefore, not only malignant epithelial cells but also other cell types may play a crucial role in producing T cell-inhibiting ligands during the onset of OC (Fig. 5d, Supplementary Data 5.2). Importantly, these cells all act through NECTIN2-TIGIT. To further validate the NECTIN2-TIGIT ligand-receptor pair as a mediator of T-cell-induced immunosuppressive microenvironment in OC, we disrupted the *NECTIN2* gene expression in human OC cell line (SKOV3) utilizing CRISPR/Cas9 targeting the coding exon 2 of the gene. The high CRISPR knock-out efficiency of *NECTIN2* was achieved (97%), with the major indel of 1-base pair insertion (94%) and a minor indel of 7-base pair deletion (3%), which was consistent with the high transduction efficiency of EGFP-mRNA in SKOV3 cells with our recent selection-free gene editing approach[46]. Additionally, the morphology of SKOV cells with *NECTIN2* knockout exhibited subtle changes, displaying a blunt phenotype (Supplementary Fig. 5a–e). To investigate the functional effect, we measured the proliferation in *NECTIN2* knockout SKOV3 cells, as well as effect on T cell proliferation/exhaustion. The *NECTIN2* knockout SKOV3 cells exhibited significant reduced proliferation rate compared to the wild-type cells ($P < 0.0001$) (Fig. 5e). Most importantly, when the activated T cells were co-cultured with *NECTIN2* knockout SKOV3 cells, we observed two-fold increase in T cell proliferation from 72 h to 240 h in two of three donors as compared to that co-cultured with wild-type SKOV3 cells (Fig. 5f, g). These results suggest that NECTIN2-TIGIT as a checkpoint mediator can inhibit T cell immune response to cancer cells. Furthermore, we performed a Kaplan-Meier survival analysis of *NECTIN2* using TCGA HGSOC data, showing that high expression of *NECTIN2* was associated with poor prognosis in patients (long-rank test, $P$ value = 0.029, Fig. 5h). In summary, our findings offer valuable insights into potential immune checkpoint therapy for OC.

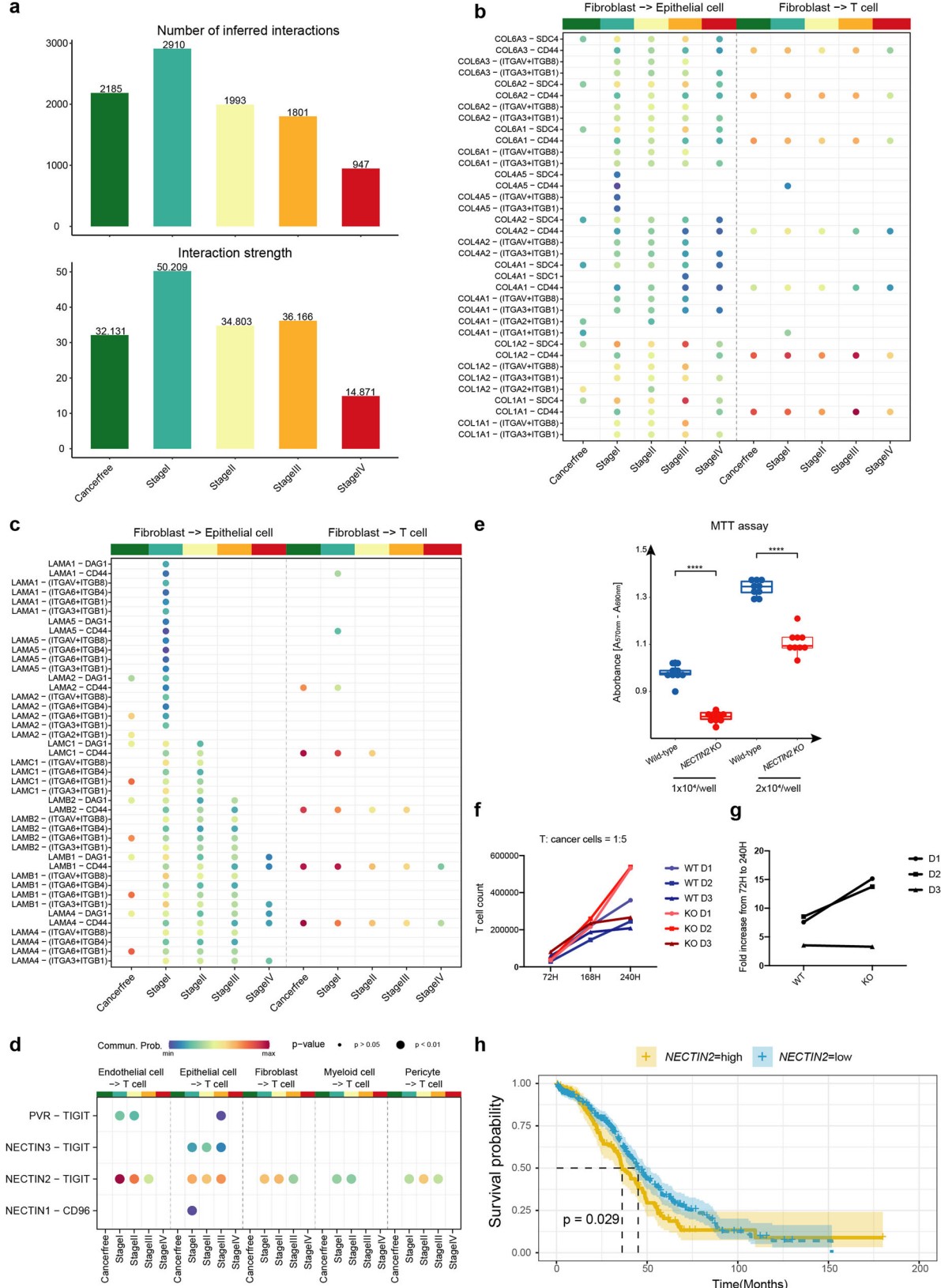

## Discussion

At present, there are few studies on the clinical stages of OC, and there are some limits such as incompleteness of stage and less metastatic tumor focal. Our study collected most of the published single-cell RNA data on OC, and endeavored to achieve whole-stage, whole-tissue types. Furthermore, we developed an open-

access database for data visualization and exploration. After completing the comprehensive single-cell atlas of OC, we annotated each epithelial cell cluster using functional gene sets, we found CNV, cell differentiation potential, cell function, and gene expression clearly change at the OC progression of stage IC1 to stage IC2. We found explicit quantity increases in T cells on both

**Fig. 5 Interactome between fibroblasts, epithelial cells, and T cells over OC progression. a** The bar chart shows the number and strength of cell interactions in ovarian cancer at different stages. **b** COLLAGEN pathway ligand-receptor cell communication probability from fibroblast to T cell. **c** LAMININ pathway ligand-receptor cell communication probability from fibroblast to T cell. **d** TIGIT ligand-receptor cell communication probability from endothelial cell, epithelial cell, fibroblast, myeloid cell, and pericyte to T cell. The color represents the communication probability and the dot size represents *p* value. **e** The inhibited cell proliferation by MTT assay in SKOV3 cells with *NECTIN2* knock out using CRISPR/Cas9 ($n = 9$ technical replicates). **f** Measurement of T cell proliferation when it was co-cultured with *NECTIN2* knock out SKOV3 cells and wilt type SKOV3 cells after 72 h, 168 h, and 240 h. T cells were derived from three individuals. **g** Fold changes of T cell proliferation when co-culturing with *NECTIN2* knock out cells compared to wild-type SKOV3 cells. **h** *NECTIN2*-high and *NECTIN2*-low Kaplan-Meier curves of patients from the TCGA cohort. *P* values were calculated by a log rank test.

ovarian tumors and metastatic tumors, mostly CD8+ exhausted T cells and Tregs. It suggested the immunosuppressive TME of OC and similarities in the immune components between metastatic and primary tumors. In addition, we explored the cell interaction in OC and discovered that fibroblasts possessed a high number of interactions and interaction strength with T cells and epithelial cells. Therefore, we also performed a subpopulation analysis of fibroblasts, uncovering an unnoticed CAF subpopulation in OC, called antigen-presenting CAF. We also found that ECM functions in myCAF had significantly enhanced since stage IC2. In our data, signaling communication between fibroblasts and epithelial cells mainly via COLLAGEN and LAMINI pathways, these molecules had reported playing important roles in tumor metastasis[47], suggesting that myCAF may be promoting tumor metastasis by COLLAGEN and LAMININ pathways. NECTIN2 and TIGIT are the most widely expressed immune checkpoint-related ligand-receptor pairs.

At the early stage of OC progress, the accuracy of existing clinical detection methods is low, and 75% of patients start showing clinical symptoms until the advanced stage[6]. Examining molecular changes in the early stages of OC may contribute to OC screening. Whole-stage data on ovarian tumors was a major feature of our study. From stage IC1 to stage IC2, clear changes occurred at the cellular and molecular levels including epithelial cell differentiation potential significantly increased, the tumor cell function such as apoptosis, DNA damage, and metastasis function evidently enhanced; CD8+ exhausted T cells and Treg cell proportions were distinctly increased, and corresponding genes expression level were up-regulated; the expression of ECM-related genes in myCAF were also up-regulated. These results all suggested changes in the microenvironment of stage IC2 ovarian tumors, and stage IC1 to stage IC2 is a critical period of OC carcinogenesis.

The immunosuppressive nature of the TME may be due to the induction of inhibitory immune checkpoints and T cell exhaustion. In this study, we observed a highly immunosuppressive microenvironment in HGSOC and metastatic tumor foci, with explicit increases in CD8+ exhausted T cells and Tregs, and impairment of effector function. *TIGIT* is the highest expressed co-suppressor receptor on CD8+ exhausted T cells. TIGIT has previously been described as an inhibitor of Tregs[48]. Antibodies targeting TIGIT have been reported to reduce the proportion of Tregs and improve the survival rate of OC mice[49]. It's reported that TIGIT and PD-L1 joint- blocking antibodies can enhance the effect of CD8 + T cells[50]. Xu et al. found that TIGIT blockade could inhibit OC tumor growth in mouse models and significantly suppressed the frequency of TIGIT + CD8 + T cells in tumors[11]. Our study identified *NECTIN2*, the ligand gene that most interacts with *TIGIT*. Nectin-2 is an adhesion molecule, has been reported as a potential antibody therapy target, suppressed OC progression in mouse models[51,52]. In line with our *NECTIN2* knockout SKOV cell results, these results collectively suggested that nectin-2 is also a potential target of immune checkpoint inhibitor therapy of OC. These results will provide valuable insights into the development of immunotherapies in OC.

In summary, in this study, we analyzed the potentially critical period of OC progression and identified some molecular changes in this period. Despite the limitations of this study including a small number of early-stage patients, further experimental work is still required to validate our study. Our study provided valuable molecular changes in the critical OC stage progression, providing further evidence for early screening and treatment of OC.

## Methods

**Data collection and processing**. 40 single cell RNA-seq literature for OC was collected and all data were downloaded as far as possible (Supplementary Data 1.1). We selected data from 10X genomics platforms to avoid batch effects across platforms. After obtaining the original single-cell expression matrix, we follow the quality control criteria described in the original article and nFeature_RNA > 500 & nFeature_RNA < 7000 & percent.mt <25 to filter the data. If the specific quality control conditions are not mentioned in the article, we follow nFeature_RNA > 500 & nFeature_RNA < 7000 & percent.mt <25 to filter (Table1).

**Data integration and clustering**. We used R package harmony[25] (version 0.1.0) to remove batch effects between samples, using parameters assay.use = "RNA", max.iter.harmony = 10. Reduction = "harmony" and dims = 1:30 were used to reduce the dimensions using the "RunUMAP" function of R package Seurat[53] (version 4.1.0). The "FindClusters" function was used to identify clusters of cells by a shared nearest neighbor modularity optimization-based clustering algorithm with resolution = 0.8.

**Consensus non-negative matrix factorization analysis**. We used Python pipeline cNMF[28] (version 1.3.4) to generate a gene expression program from epithelial cells. We used parameters n_iter = 20, num_highvar_genes = 2000, components = np.arange (3,26) to generate separate files. We selected 9 as the optimal k value and reduction = "NMF" and dims = 1:9 were used to reduce the dimensions using the "RunUMAP" function of R package Seurat (version 4.1.0). The "FindClusters" function was used to identify clusters of cells by a shared nearest neighbor modularity optimization-based clustering algorithm with resolution = 0.15.

**GO enrichment analysis**. We used the R package org.Hs.eg.db (version 3.16.0) to convert the symbol and entrezid. Then GO enrichment analysis was performed using the clusterProfiler[54] (version 4.6.0) "enrichGO" function with the parameters ont = "BP" and *p* valueCutoff = 0.05. Finally, ggplot2 (version 3.3.5) was used for visualization.

**Identification of DEGs**. We performed the "FindMarkers" function of package R Seurat to find markers (DEGs) for identity classes with logfc.threshold = 0.25, min.pct = 0.25, only.pos = T, return.thresh = 0.05.

**GSVA**. We applied GSVA[55] (version 1.46.0) for scoring analysis of functional gene sets in tumor cells and the top 10 marker genes of myCAF. We then used pheatmap (version 1.0.12) for visualization.

**Pseudo-time analysis**. We used the Monocle2[56] (version 2.22.0) to explore the underlying changes in epithelial cell function and identify potential lineage differentiation. We provide the original UMI count gene cell matrix as input to Monocle, and then use the "newCellDataSet" function to create a Monocle object with lowerDetectionLimit = 0.1, expressionFamily = negbinomial.size(). Then, we applied "reduceDimension" to reduce the data dimension with method = "DDRTree" and max_components = 2. We performed "diferentialGeneTest" to identify genes that differed significantly over time.

We used monocle3 (version 1.2.9) to explore the differentiation trajectory of CD8T cell subtypes. We use the "learn_graph" function to learn the principal graph from the reduced dimension space using reversed graph embedding. After that, we defined the root node as CD8 naive T cells and assigned pseudo-time values to the cells according to the projection on the master graph that the cells learned in the "learn_graph" function. Next, we use "graph_test" to test genes for differential expression based on the low dimensional embedding and the principal graph. Finally, we use "plot_cells" for visualization.

**Single-cell copy number variation analysis**. We used inferCNV (version 1.10.1) R package to estimate CNV level in malignant epithelial cell. We used an approach described on the website tutorial (https://github.com/broadinstitute/inferCNV), used parameter –cutoff 0.1, and selected average expression as reference.

**Assessment of cell differentiation potential**. R package SCENT[57] (version 1.0.3) was used to estimate the epithelial cell differentiation potency. We used the default parameters described on the website tutorial (https://github.com/aet21/SCENT/blob/master/inst/doc/SCENT.Rmd) to calculate cell entropy.

**Identifying time-dependent transcriptional program in epithelial cell, T cell, and fibroblast**. We used Mfuzz[30] (version 2.54.0) to identify time-dependent transcriptional programs in epithelial cell, T cell, and fibroblast. First, the average expression of each highly variable gene was calculated for each stage. Next, we used the "filter.std(min.std = 0)" and "standardize()" functions for preprocessing according to the tutorial. Then, "mestimate()" functions were performed to estimate the most appropriate value of m. Finally, we clustered the genes into 20 different expression programs.

**Single cell gene set scoring**. We used the "AddModuleScore" function of the R package Seurat to calculate the average expression levels of each program on a single cell level. Gene sets were obtained by literature or GO database collation (http://amigo.geneontology.org/amigo/term/). Other parameters are used by default.

**Survival analysis**. The R package cgdsr (version 1.3.0) was used to link to the cBioPortal database (http://www.cbioportal.org/) and download the TCGA-OV data (ov_tcga_pan_can_atlas_2018). The expression matrix of target genes was obtained by the "getProfileData" function. The "getClinicalData" function gets the clinical information of the case. The "surv_cutpoint" function of R package survminer (0.4.9) was then used to determine the optimal cutpoint of variables. Finally, "survfit" was used for survival analysis, and "ggsurvplot" for visualization.

**Ligand-receptor interaction analysis**. We used CellChat[58] (version 1.5.0) to perform ligand-receptor interaction analysis. ScRNA-seq gene expression matrix was used as input data. "ChatDB.human" was set for our database. We ran "computeCommunProb" to infer the probability and strength of cell-cell communication. Merge cellchat objects for each stage using the "mergeCellChat" function. We compare the total number of interactions and interaction strength using the "compareInteractions" function. We used "netVisual_bubble" to visualize ligand-receptor strength for merged objects.

**Guide RNA design and oligonucleotide synthesis**. The CRISPR-SpCas9 sgRNA targeting *NECTIN2* was designed with the online CRISPR design tool "CRISPRon" (https://rth.dk/resources/crispr/crispron/)[59]. The sgRNA was synthesized by IDT (Integrated DNA Technologies), and the oligonucleotides for PCR were ordered from Merck KGaA, Darmstadt, Germany. Sequences for all oligonucleotides can be found in Supplementary Data 6.

**Nucleofection of SKOV3 cells with EGFP-mRNA and *NECTIN2* CRISPR-Cas9**. EGFP mRNA was generated by in vitro transcription (IVT)[46].

In brief, the EGFP plasmid was linearized by a reaction mixture containing 15 µL nuclease-free water (Thermo Scientific), 2 µL 10X Fast Digest Green Buffer (Thermo Scientific), 2 µL EGFP plasmid, and 2 µL Fast Digest restriction enzyme BbsI (Thermo Scientific) at 37 °C for 3 h. Next, the linearized EGFP was visualized on a 1% agarose gel and purified using a NucleoSpin Gel and PCR Clean-up (Macherey) following the manufacturer's protocol. Finally, IVT was carried out using a MEGAscript kit (Thermo Scientific) following the manufacturer's protocol, with an addition of 3 µL CleanCap AG (6 mM) (TriLink Biotechnologies) to enhance the stability of the mRNA and its translation efficiency. The concentration of EGFP mRNA was measured by a Nanodrop and stored at −20 °C. The synthesized sgRNAs (IDT) were dissolved to be 3.2 µg/µL in nuclease-free water mixed by vortexing and stored at −20 °C. Preparations for the nucleofection: The cells were gently washed twice in PBS, centrifuged at 500 x g for 5 min, and resuspended in 60 µL OptiMem (Gibco) with 200,000 cells per group for three groups. The cells in three groups were prepared with ① 1 µg EGFP-mRNA, ② RNP complex, 1 µL NECTIN2 sgRNA (3.2 µg/µL) + 6 µL spCas9 Nuclease V3 (IDT), ③ control group, and kept at room temperature for 10–60 min. The cells were then nucleofected by the 4D-Nucleofector X Unit (Lonza) with the nucleofection program: CM138. After nucleofection, 150 µL prewarmed DMEM medium with supplements was added to each well in the nucelocuvette. Finally, the nucleofected SKOV3 was seeded in the prepared 24-well plate with a total volume of 500 µL culturing medium and incubated for 24 h in a 5% CO2 atmosphere at 37 °C. The EGFP fluorescence was visualized by a ZOE Fluorescent Cell Imager (BIO-RAD) fluorescent microscope to ensure efficient nucleofection after 24 h of incubation.

**PCR, Sanger sequencing, and ICE analysis**. Genomic DNA of cells (*NECTIN2* knockout SKOV3 and wild type SKOV3 cells) were extracted with a 200 µL homemade lysis master mix 905 lysis buffer CS (KCL 50 mM, MgCl2 1.5 mM, Tris/HCL pH8.5 10 mM) + 25 µL Tween20, 20% (Sigma) + 50 µL NP-40 Surfact-Amps Detergent (Thermo Scientific) + 20 µL proteinase K (Roche), and incubated at 65 °C for 30 min and 95 °C for 15 min. The cell lysate was stored at stored at −20 °C for PCR. The PCR reactions were conducted with

DreamTaq DNA polymerase (Thermo Fisher). The PCR products were purified with a Gel and PCR Clean-up Kit (Macherey-Nagel) by following the manufacturer's protocol. The products were prepared for Sanger sequencing. The ICE analysis was then used to evaluate the total indel profiles and knock-out efficiency (https://ice.synthego.com/).

**Cell Proliferation by MTT assay.** The alterations of cell proliferation for *NECTIN2* knock-out SKOV3 cells compared to wild-type SKOV3 cells were measured with Cell Proliferation Kit I (MTT, Roche) according to the manufacturer's instructions. In brief, two groups of wild-type and Nectin 2 knock-out SKOV3 cells were seeded with the initial cell numbers of $1 \times 10^4$ and $2 \times 10^4$ for cell proliferation. After 24 h of culturing, the purple formazan crystals were checked for total solubilization, and the spectrophotometrical absorbance of the samples was measured using a spectraMax ID3 plate reader.

**PBMC and T cell isolation.** Peripheral blood mononuclear cells were isolated from de-identified buffy coats obtained from healthy adult donors from the Aarhus University Hospital Blood Bank by a Ficoll-Paque Plus (Cytiva, Thermo Fisher Scientific) density gradient. Primary human CD3 + T cells were subsequently purified by immunomagnetic negative enrichment with the EasySep Human T Cell Isolation Kit (STEMCELL Technologies) according to the manufacturer's instructions.

**Cell culture.** SKOV3 cells, an ovarian adenocarcinoma cell line, were cultured in DMEM, high glucose medium (Gibco) supplemented with 10% Fetal Bovine Serum (Merck), and penicillin-streptomycin (100 units penicillin and 0.1 mg streptomycin/mL). Primary human T cells were cultured in X-vivo 15 media (Lonza) supplemented with 5% human serum (Sigma-Aldrich), 100 IU/mL IL-2 (PeproTech), and 10 ng/mL IL-7 (PeproTech). T cells were activated for 3 days with Human T-Activator CD3/CD8 Dynabeads (Thermo Fisher Scientific) at a 1:1 cell-to-bead ratio before experiments.

**Co-culture and T cell proliferation assay.** Primary human T cells were co-cultured with either WT or *NECTIN2* knockout SKOV3 cells at a 1:5 ratio in X-vivo 15 media. Additional media were added as required during the co-culture period. T cell proliferation was determined at 72H, 168H, and 240H using CountBright Absolute Counting beads (Thermo Fisher Scientific) according to the manufacturer′s instructions with slight modifications. Briefly, 60 µL cell suspension containing primary human T cells were mixed with 5 µL counting beads, 2.5 µL 50 µg/mL propidium iodide (PI), and adjusted to a total volume of 100 µL with FACS buffer (PBS, 2% FBS, 2 mM EDTA). Cells were incubated for 5 min at room temperature in the dark and analyzed on a 4-laser Cytoflex S flow cytometer (Beckman Coulter) with a fixed stop condition at 60 µL using the CytExpert acquisition software. The live cell density was calculated as follows: live cells/mL = (number of live cells counted/number of beads counted) * (number of beads added to the sample/sample volume), which were used to determine the total cell number based on the total culture volume.

**Statistics and reproducibility.** Statistical analyses were performed through the R package ggpubr (version 0.4.0). The Wilcoxon rank-sum test was used to test the significance of most cases. Statistical significance was defined as $p < 0.05$ (*$p < 0.05$, **$p < 0.01$, ***$p < 0.001$, ****$p < 0.0001$; ns, not significant). Survival analysis was analyzed using the log-rank test. The experiments were conducted with at least three repetitions.

**Reporting summary.** Further information on research design is available in the Nature Portfolio Reporting Summary linked to this article.

## Data availability
The integrated single cell transcriptome data used for the analysis in this study is available in this open access database with the url: https://dreamapp.biomed.au.dk/OvaryCancer_DB/. Source data underlying Figs. 1d, 2c, 2h, 4d, 5b-d are presented in Supplementary Data 1.2, 1.3, 2.1, 2.5, 4.2, 5.1, 5.2. Other data are available upon reasonable request to the corresponding authors.

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

## Acknowledgements

We thank the China National GeneBank DataBase for providing computational resource for the storage, processing, and analysis of the single cell RNA sequencing data. We would like to thank Trine Skov Petersen and Yan Zhou for technical assistance. This project is partially supported by the Qingdao-Europe Advanced Institute for Life Sciences. Y.L. is supported by the Novo Nordisk Foundation (NNF21OC0072031 and NNF21OC0068988) and the Lundbeck Foundation (R396-2022-350). Collaboration between Y.L. and R.O.B is based on supports from COST Action Gene Editing for the treatment of Human Diseases, CA21113, supported by COST (European Cooperation in Science and Technology).

## Author contributions

Conceptualization: C.C., La.L., F.W., Y.L. Methodology: C.C., La.L.. Software: C.C., La.L., L.L. Validation: W.W., W.Z., C.S, Formal analysis: C.C., La.L., W.W., W.Z., C.S. Investigation: C.C., La.L., W.W., W.Z., C.S., H.L. Experimental validation: F.W., N.S.M., R.O.B., Y.L. Writing - Original Draft: C.C., La.L. Writing - Review & Editing: C.C., La.L., W.W., W.Z., C.S, H.L, L.L., F.W., Y.L. Visualization: C.C., La.L., F.W. Supervision: F.W., Y.L. Funding acquisition: Y.L.

## Competing interests

The authors declare no competing interests.

## Ethical approval

The inclusion of collaborators and authors to all the research and analytic work in this study is conducted according to the Global Code of Conduct.
