## [Peer Review File · Communications Biology]

Reviewers' comments:

Reviewer #1 (Remarks to the Author):

The manuscript by Luo et al details efforts to reveal the cellular composition of human ovarian cancer from the published scRNA-seq data. Tumors from 84 patients were subjected to transcriptional analysis with single cell resolution. Cataloging of each major cell type subsequently revealed multiple subsets with distinct functional properties, as deduced by gene expression patterns. The study holds limited novelty. At best, this is very descriptive work and complement other published work.

1. The whole manuscript is exclusively built on speculative inferences based on differentially expressed genes and different tumor stages. Most of the information presented is already known. What new information did the reanalyzed scRNA-seq provide?
2. The description about T cells in introduction section should be rewritten to clearly justify the analyses carried out.
3. Each specific subcluster of the tumor cell types should be renamed. For example, marker gene A-Epi 1, marker gene B-Epi 2, and so on.
4. The authors state that "CD8+ exhausted T cells and Tregs significantly increased after stage IC2", however, as shown in figure 3E, CD8+ exhausted T cells and CD8+ proliferating T cells were decreased across the tumor progression from IC2 to IV. The authors should validate the findings in different tumor stages of ovarian cancer.
5. Figure 4F, please add all the p value labels.
6. The accompanying survival analyze in Figure 5 is correlative at best, and separation of the curves is not obvious. The X axis label is not complete.
7. Statistical analysis should be described in detail.
8. Validation of the scRNA data at the protein or cellular level in situ is conspicuously missing.

Reviewer #2 (Remarks to the Author):

I reviewed the manuscript titled "Single cell analysis of epithelial, immune, and stromal signatures and interactions in human ovarian cancer" by Chai and colleagues. The authors performed integrative analysis of single cell transcriptome data of ovarian tumor patients in multiple public datasets and identified the heterogeneous epithelial-immune-stromal cellular compartments and their interactions in ovarian cancers. They observed distinct T cell subtypes in different stages of the disease along with antigen-presenting cancer-associated fibroblasts (CAFs), and myofibroblastic CAFs exhibiting enriched extracellular matrix signature linked to tumor progression. In silico analysis also identified NECTIN2-TIGIT ligand-receptor pair mediating T cell communication with other cell types. Methods are clearly described, results are adequately presented and described. The reported findings shed light on the cellular compartments and potential their functional aspects of ovarian cancer. The manuscript requires substantial English languish editing.

Reviewer #3 (Remarks to the Author):

Chai et al. report in this study a single cell atlas analysis of epithelial, immune, and stromal signatures and interactions in human ovarian cancer. This could be a great resource for future researchers. Overall, it is a good paper, the following comments were made during the review of this paper:

1. The authors performed cell interaction analysis with CellChat using all data from different sources, whether the entire harmony integrated data as one, or process separately for each sample? I have

some concerns that using harmony integrated data as one might bring unknown consequences when establishing connections between cell types between different samples. See a discussion of the technique here, <https://github.com/sqjin/CellChat/issues/127>.

2. For the Fig 5-E, I would like to see if the counterpart of NECTIN2, TIGIT, its survival performance in TCGA OV, and their correlation at gene expression levels (scatter plot), which will provide more evidence for the conclusion.

Reviewer #1 (Remarks to the Author):

The manuscript by Luo et al details efforts to reveal the cellular composition of human ovarian cancer from the published scRNA-seq data. Tumors from 84 patients were subjected to transcriptional analysis with single cell resolution. Cataloging of each major cell type subsequently revealed multiple subsets with distinct functional properties, as deduced by gene expression patterns. The study holds limited novelty. At best, this is very descriptive work and complement other published work.

Response: We appreciate the reviewer's critical comments and suggestions to the improvement of the work. With the increasing amount of single cell RNA sequencing data, the value of integrating all the published dataset on a focus complication, in our case for OC, is valuable for creating valuable resource for the field of research. We have thoroughly considered all the suggestions and include more functional validation on the cellular levels to validate our finding. Using CRISPR gene editing, we knockout the NECTIN2 in human ovarian cancer cells and validated that it affects cancer cell proliferation and T cell exhaustion.

1. The whole manuscript is exclusively built on speculative inferences based on differentially expressed genes and different tumor stages. Most of the information presented is already known. What new information did the reanalyzed scRNA-seq provide?

Response: Thank you very much for your comments. In the early stages of ovarian cancer (OC), stages I and II of The International Federation of Gynecology and Obstetrics (FIGO), patients have a higher five-year survival rate. However, the symptoms of OC are not specific, leading to the majority of cases diagnosed in late stages (FIGO stages III and IV) (< 70%), while tumor metastasis occurs and five-year survival drops below 30% [1, 2]. Due to the relative ease of obtaining samples from late-stage patients, most studies have focused on late-stage patients [3, 4], metastatic tumors [4, 5], and mechanisms of metastasis [6], and there are very few studies on the early stage of ovarian cancer. Proteomic and bulk transcriptomic studies encompassing early-stage OC have the potential to discover early screening biomarkers for ovarian cancer [7-10]. **However, the existing studies have used the FIGO stage I-IV, which is not precise enough, and mostly focuses on the differences between stages I-II and III-IV [7], lacking research on differences of finer FIGO stage.** Single-cell analysis studies related to OC stages have revealed differences in cell types between different OC stages, such as a decrease in the attractiveness of immune cells by macrophages as the stage progresses [11]. This suggests that there are complex functional changes in ovarian cancer cells through the stage changes, but the collection of the included OC samples in this study is incomplete and the targeted cell types are not comprehensive. How do the proportions and functional changes of other cell types vary between finer stages of ovarian cancer? We hope to analyze the difference between the most **comprehensively distributed cell types** such as epithelial cells, T cells, and cancer-associated fibroblasts (CAFs) in ovarian cancer through the stage progress by collecting more comprehensive single-cell data of finer stages. We aim to identify key inflection points at the single-cell level and elucidate the changes in cell function and cell proportions at these inflection points. Compared to previous ovarian cancer research, our study provides the new information, including but not limited to:

1. We have collected as comprehensive as possible single-cell transcriptomic data of OC with fine

FIGO staging, and integrated and annotated these data. This will be regarded as an important resource of database generation for the field.

2. Significant changes in epithelial cells, T cells, and fibroblasts occurred between stages IC1-IC2, indicating a cellular remodeling in the early tumor progression.
3. Antigen-presenting cancer-associated fibroblasts (CAFs) are found in ovarian cancer, which is a very interesting observation. We speculate that these subtype of CAFs are associated with the anti-tumor immunity in the cancer progression.
4. The most widely expressed T-cell exhaustion-related ligand pairs in ovarian cancer were identified from the perspective of cell interactions, particularly for the NECTIN2-TIGIT validated with the new cellular experiment.
5. Collected the most comprehensive transcriptome data available on single cells of ovarian cancer and produced an online website for researchers to use.

2. The description about T cells in introduction section should be rewritten to clearly justify the analyses carried out.

Response: Thanks for your suggestion. We have revised the description of T cells in the introduction section.

3. Each specific subcluster of the tumor cell types should be renamed. For example, marker gene A-Epi 1, marker gene B-Epi 2, and so on.

Response: Thanks for your kind reminder. We have renamed the tumor cell subtype.

4. The authors state that "CD8+ exhausted T cells and Tregs significantly increased after stage IC2", however, as shown in figure 3E, CD8+ exhausted T cells and CD8+ proliferating T cells were decreased across the tumor progression from IC2 to IV. The authors should validate the findings in different tumor stages of ovarian cancer.

Response: Thank you very much for pointing out the problem in our description. What we wanted to express here is that "CD8+ exhausted T cells and Tregs significantly increased during IC2-IVB compared to Cancer-free IC1", and we also made the corresponding modifications to the manuscript. Here we highlight the finding that T cells in ovarian cancer have significant immunosuppressive enhancement in IC1-IC2 at the cellular level, and the gene expression heat map of fig3E demonstrates this finding at the gene level. Moreover, in Figure S3B-C, we calculated the Treg differentiation score and Exhaustion score of each phase, and at stage IC1-IC2, the two scores also increased significantly, which also verified our findings.

In the table below, we summarize the number of patients and T cell counts in the ovaries at different FIGO stages, and our conclusions are all supported by large sample sizes, which is important.

Stages	Cancer-free	IA	IC1	IC2	IIB	IIIB	IIIC	IV	IVA	IVB
--------	-------------	----	-----	-----	-----	------	------	----	-----	-----

Patients number	5	1	1	3	3	1	11	2	2	1
Cell number	32392	409	206	4490	3281	1096	22478	2770	10876	4484

Furthermore, according to the most recent FIGO Ovarian Cancer Staging effective January 2018 (<https://www.cancer.org/cancer/types/ovarian-cancer/detection-diagnosis-staging/staging.html>), the clinical characteristics of IC1 are as follows: The tissue (capsule) surrounding the tumor broke during surgery, which could allow cancer cells to leak into the abdomen and pelvis (called surgical spill). The clinical characteristics of IC2 are as follows: Cancer is on the outer surface of at least one of the ovaries or fallopian tubes or the capsule (tissue surrounding the tumor) has ruptured (burst) before surgery (which could allow cancer cells to spill into the abdomen and pelvis). This indicates that there is a change in clinical characteristics from IC1 to IC2, with the presence of cancer cells on the outer surface of the ovaries. Our findings reveal a significant alteration in the proportion and molecular level of T cells in the ovaries from IC1 to IC2. This is likely correlated with the clinical characteristics. In other words, when IC2 is reached, cancer cells start to gain an advantage and occupy the ovarian surface, leading to a significant depletion of T cells.

5. Figure 4F, please add all the p value labels.

Response: Thanks for your valuable suggestions. We have added all p value labels in Figure 4F.

6. The accompanying survival analyze in Figure 5 is correlative at best, and separation of the curves is not obvious. The X axis label is not complete.

Response: Thanks for pointing out this problem. We have revised the relevant description and added the X-axis label.

7. Statistical analysis should be described in detail.

Response: Thank you for the valuable and helpful comments. We have added the corresponding description in the revised methods, as well as in the figures.

8. Validation of the scRNA data at the protein or cellular level in situ is conspicuously missing.

Response: We would like to express our great appreciation for your constructive comments. Regarding the validation of the markers at the protein levels, we have provided independent data from the protein atlas for the response letter here, as well as data supported by other published studies. Instead, in this revision, we focus on validating the key finding of the NECTIN2-TIGIT connectome in OC progression. To validate the functions of NECTIN2-TIGIT ligand-receptor pair mediating T cell immunity and cancer cell progression in OC. We generated NECTIN2 knockout human ovarian cancer cell line (SKOV3) using CRISPR/Cas9 and performed functional assays. New data are provided in supplementary Figure 5 and Figure 5.

We evaluated the cell proliferation in the NECTIN2 knockout SKOV3 cells. Our results show that

NECTIN2 knockout significantly inhibits SKOV3 proliferation.

Furthermore, to investigate whether NECTIN2 affects T cell proliferation and exhaustion. We cultured activated T cells with NECTIN2 knock-out SKOV3 or with WT SKOV3 cells. The result showed that the proliferation of the activated T cells was significantly increased when co-cultured with the NECTIN2 KO SKOV3 in two of three T cell donors, as compared to co-cultured with WT SKOV3 cells. The other T cell donor was not responding, also for the WT cells. It indicates that expression of NECTIN2 by cancer cells negatively affect activated T cell proliferation, and thus contributing to T cell exhaustion in TME.

In addition, since most of the analysis in this study was aimed at the cellular and gene level changes of ovarian cancer epithelial cells, T cells, and fibroblasts in different periods, it was difficult to find patients of all time for the verification of cell or protein levels, especially in the early stage, it was very difficult to collect patients. We therefore validated our stage-varying genes (Fig2-4) by analyzing published bulk transcriptome data (GSE44104). The results in Figure 5 were verified by querying the human protein map and published literature.

As illustrated in the figure below, most of the genes mentioned in **Figure 2-4** are significant increase in stage II compared with stage I. The stage information for this data is only stages I, II, III, and IV, which is a pity.

2. The expression of NECTIN2 was verified with data from the human protein atlas (<https://www.proteinatlas.org/ENSG00000130202-NECTIN2/pathology/ovarian+cancer#>). NECTIN2 antibody staining of ovarian cancer tissues showed that NECTIN2 was highly expressed in almost all cell types in ovarian cancer tissues, which was consistent with our findings in Fig 5D.

3. Expression of the genes is also supported by published studies

① Real-time quantitative polymerase chain reaction (qRT-PCR) and western blot showed that nectin-2 was overexpressed in ovarian cancer cells [12].

Nectin-2 expression was examined using qRT-PCR (A) and western blot (B). Daudi cells were used as a *nectin-2* negative cell line.

② Nectin-2 protein is overexpressed in ovarian cancer tissues [12].

Paraffin-embedded tissue sections were stained with anti-Nectin-2 poAb as described in Methods. E, normal ovarian tissue, and F–H, ovarian serous carcinoma tissues.

4. Data from the Ovarian Cancer Research Alliance online website (<https://www.ovara.net>) were used to verify genes mentioned in Fig5. This OCRA database compiles resources by combining data generated by research consortiums with published datasets. These included transcriptomic (RNA-Seq), proteomic (LC-MS/MS), and secretomic data for tumors and tumor-associated host cells, as well as overall and relapse-free survival data.

tumor cells (TU), macrophages (TAM) and T cells (TAT)

As shown with the figures above, NECTIN2 is highly expressed in tumor cells in transcriptomic (RNA-Seq), proteomic (LC-MS/MS) and secretomic data.

We also used public data to explore association between survival rate of COLLAGEN and LAMININ levels. The results showed that most of COLLAGEN and LAMININ showed shorter survival (hazard ratio >1), which further verifies our inference in Fig5B-C: “fibroblast cells may interact with epithelial cells through COLLAGEN and LAMININ pathways in the tumor stage, leading to tumor cell metastasis.”

Survival association (negative=beneficial)

Survival association (negative=beneficial)

Survival association (negative=beneficial)

Survival association (negative=beneficial)

Association of transcriptome and survival data:

1.RFS associated with TCGA microarray data.

2.KM-Plotter RNA microarray / relapse-free survival (RFS) data for HGSC (10.1530/ERC-11-0329).

3.PRECOG RNA-Seq / overall survival data for ovarian cancer (10.1038/nm.3909).

TCGA, KMP: data split at best fitting quantile; PRECOG: data split at median.

A z-zcore of >1.96 or <-1.96 corresponds to a logrank p-value of <0.05.

Positive z-scores indicate a hazard ratio >1 (shorter survival); negative z-scores indicate a hazard ratio <1 (longer survival).

Reviewer #2 (Remarks to the Author):

I reviewed the manuscript titled “Single cell analysis of epithelial, immune, and stromal signatures and interactions in human ovarian cancer” by Chai and colleagues. The authors performed integrative analysis of single cell transcriptome data of ovarian tumor patients in multiple public datasets and identified the heterogeneous epithelial-immune-stromal cellular compartments and their interactions in ovarian cancers. They observed distinct T cell subtypes in different stages of the disease along with antigen-presenting cancer-associated fibroblasts (CAFs), and myofibroblastic CAFs exhibiting enriched extracellular matrix signature linked to tumor progression. In silico analysis also identified NECTIN2-TIGIT ligand-receptor pair mediating T cell communication with other cell types. Methods are clearly described, results are adequately presented and described. The reported findings shed light on the cellular compartments and potential their functional aspects of ovarian cancer. The manuscript requires substantial English languish editing.

Response: We thank the reviewer for the recommendation of our study for publication. In this revision, we have further polished the language to be clear and precise for the content and findings. Furthermore, we have carried out functional experiments to further validate the key findings from the study. We strongly believe that the manuscript has been substantially improved and will provide valuable resource and findings regarding the molecular and cellular mechanisms in the progression of OC.

Reviewer #3 (Remarks to the Author):

Chai et al. report in this study a single cell atlas analysis of epithelial, immune, and stromal signatures and interactions in human ovarian cancer. This could be a great resource for future researchers. Overall, it is a good paper, the following comments were made during the review of this paper:

Response: Thank you for the great comments. We have in the revision version of the manuscript thoroughly address them. These are very help for the improvement of our analysis and the manuscript.

1. The authors performed cell interaction analysis with CellChat using all data from different sources, whether the entire harmony integrated data as one, or process separately for each sample? I have some concerns that using harmony integrated data as one might bring unknown consequences when establishing connections between cell types between different samples. See a discussion of the technique here, <https://github.com/sqjin/CellChat/issues/127>.

Response: Thanks for your suggestion. The CellChat analysis in Figure5 belongs to comparative analysis. We subset the whole harmony integrated data into one subset, then subset the data of each stage, then run cellchat analysis of the data of each stage, and finally merge the data of five stages together for comparison. Your concerns are valid, as CellChat author sqjin replied, we did the same normalization for each stage dataset (Seurat::NormalizeData(normalization.method = "LogNormalize", scale.factor = 10000), then re-run the cellchat analysis and got exactly the same results as before. After the same normalization treatment for each sample, cellchat analysis was re-run, and the results were still consistent. This is because Seurat normalizes the original expression (counts matrix) of each cell, so whether it normalizes the stage or the sample, the expression matrix

will be the same, so the same result will be obtained. Thank you for bringing this to our attention.

2. For the Fig 5-E, I would like to see if the counterpart of NECTIN2, TIGIT, its survival performance in TCGA OV, and their correlation at gene expression levels (scatter plot), which will provide more evidence for the conclusion.

Response: Thank you for the valuable and helpful comments. For Fig5E, we plotted the Kaplan-Meier survival curve of TIGIT, and the result was opposite to NECTIN2, that is, the lower the expression of TIGIT, the shorter the survival time of ovarian cancer patients. To verify the accuracy of the results, we at <http://kmplot.com/analysis/index.php?p=service&cancer=ovar#> have done TIGIT survival curve analysis, the result is consistent. This is odd, but interesting, so we conducted literature research. TIGIT is reported to be protective, it through negative regulating NK liver cell crosstalk to promote liver regeneration [13]. TIGIT + NK cells showed higher cytotoxicity ability and maturity [14]. To verify this, we downloaded the bulk transcriptome data of ovarian cancer (GSE44104), grouped according to TIGIT expression, and found that the group with high TIGIT expression was indeed also highly expressed in NK cell cytotoxic genes. Analysis of the sample data of 1286 breast cancer TIGIT [15] associated with better prognosis of breast cancer. Not only breast cancer, TIGIT has been reported to have Skin Cutaneous Melanoma, Rectum adenocarcinoma, Uterine Corpus Endometrial Carcinoma, Adrenocortical carcinoma, Breast invasive carcinoma, Cervical squamous cell carcinoma and endocervical adenocarcinoma, Prostate adenocarcinoma, Head and Neck squamous cell carcinoma, colorectal cancer in the show and negatively correlated with overall survival (OS) [16-19]. In conclusion, these results suggest that TIGIT activity may vary between tumors and relative to normal tissues, and that it may have different roles due to cooperation with other immune molecules to regulate the immune microenvironment.

We also calculated the Person correlation of NECTIN2-TIGIT at the level of gene expression (scatter plot, where a dot represents a cell, X axis is the expression of TIGIT, Y axis is the expression of NECTIN2). We found almost no correlation between NECTIN2 and TIGIT at the gene expression level ($R=-0.19$). This is reasonable because NECTIN2 is expressed in endothelial cell, Epithelial cell, Fibroblast, Myeloid cell, pericyte, while TIGIT is mainly expressed in T cells. In other words, cells expressing NECTIN2 do not express TIGIT, and cells expressing TIGIT do not express NECTIN2, so it is not surprising that there is not correlation between them.

References:

1. Nebgen, D.R., K.H. Lu, and R.C. Bast, Jr., *Novel Approaches to Ovarian Cancer Screening*. *Curr Oncol Rep*, 2019. **21**(8): p. 75.
2. Stewart, C., C. Ralyea, and S. Lockwood, *Ovarian Cancer: An Integrated Review*. *Semin Oncol Nurs*, 2019. **35**(2): p. 151-156.
3. Geistlinger, L., et al., *Multitomic Analysis of Subtype Evolution and Heterogeneity in High-Grade Serous Ovarian Carcinoma*. *Cancer Res*, 2020. **80**(20): p. 4335-4345.
4. Zhang, K., et al., *Longitudinal single-cell RNA-seq analysis reveals stress-promoted chemoresistance in metastatic ovarian cancer*. *Sci Adv*, 2022. **8**(8): p. eabm1831.
5. Deng, Y., et al., *Single-Cell RNA-Sequencing Atlas Reveals the Tumor Microenvironment of Metastatic High-Grade Serous Ovarian Carcinoma*. *Front Immunol*, 2022. **13**: p. 923194.
6. Kan, T., et al., *Single-cell RNA-seq recognized the initiator of epithelial ovarian cancer recurrence*. *Oncogene*, 2022. **41**(6): p. 895-906.
7. Barnabas, G.D., et al., *Microvesicle Proteomic Profiling of Uterine Liquid Biopsy for Ovarian Cancer Early Detection*. *Mol Cell Proteomics*, 2019. **18**(5): p. 865-875.
8. Wang, J., et al., *Proteomic studies of early-stage and advanced ovarian cancer patients*. *Gynecol Oncol*, 2008. **111**(1): p. 111-9.
9. Nossov, V., et al., *The early detection of ovarian cancer: from traditional methods to proteomics. Can we really do better than serum CA-125?* *Am J Obstet Gynecol*, 2008. **199**(3): p. 215-23.
10. Hulstaert, E., et al., *Candidate RNA biomarkers in biofluids for early diagnosis of ovarian cancer: A systematic review*. *Gynecol Oncol*, 2021. **160**(2): p. 633-642.
11. Xu, J., et al., *Single-Cell RNA Sequencing Reveals the Tissue Architecture in Human High-Grade Serous Ovarian Cancer*. *Clin Cancer Res*, 2022. **28**(16): p. 3590-3602.
12. Sim, Y.H., et al., *A Novel Antibody-Drug Conjugate Targeting Nectin-2 Suppresses Ovarian Cancer Progression in Mouse Xenograft Models*. *Int J Mol Sci*, 2022. **23**(20).
13. Bi, J., et al., *TIGIT safeguards liver regeneration through regulating natural killer cell-hepatocyte crosstalk*. *Hepatology*, 2014. **60**(4): p. 1389-98.
14. Chauvin, J.M. and H.M. Zarour, *TIGIT in cancer immunotherapy*. *J Immunother Cancer*, 2020. **8**(2).
15. Fang, J., et al., *Prognostic value of immune checkpoint molecules in breast cancer*. *Biosci Rep*, 2020. **40**(7).
16. Giampietri, C., et al., *Analysis of gene expression levels and their impact on survival in 31 cancer-types patients identifies novel prognostic markers and suggests unexplored immunotherapy treatment options in a wide range of malignancies*. *J Transl Med*, 2022. **20**(1): p. 467.

17. Liang, R., et al., *TIGIT promotes CD8(+)T cells exhaustion and predicts poor prognosis of colorectal cancer*. *Cancer Immunol Immunother*, 2021. **70**(10): p. 2781-2793.
18. Murakami, D., et al., *Prognostic value of CD155/TIGIT expression in patients with colorectal cancer*. *PLoS One*, 2022. **17**(3): p. e0265908.
19. Tang, L., et al., *Expression and Clinical Significance of TIGIT in Primary Breast Cancer*. *Int J Gen Med*, 2023. **16**: p. 2405-2417.

REVIEWERS' COMMENTS:

Reviewer #1 (Remarks to the Author):

I carefully reviewed the revised manuscript. The authors have revised the whole manuscript based on the comments from the previous review. I recommend accepting this article.

Reviewer #2 (Remarks to the Author):

The authors adequately addressed my concerns.

REVIEWERS' COMMENTS:

Reviewer #1 (Remarks to the Author):

I carefully reviewed the revised manuscript. The authors have revised the whole manuscript based on the comments from the previous review. I recommend accepting this article.

Reviewer #2 (Remarks to the Author):

The authors adequately addressed my concerns.

Response:

We are glad to hear that both reviewers are pleased about the revision and their recommendations of our study to be published in Communications Biology.

Thank you again for all the highly valuable suggestions and comments for improving our study and the manuscript. All these contributions are highly valuable for us, as well as for the scientific communication.